# Antimicrobial Activities of Different Fractions from Mucus of the Garden Snail *Cornu aspersum*

**DOI:** 10.3390/biomedicines8090315

**Published:** 2020-08-28

**Authors:** Aleksandar Dolashki, Lyudmila Velkova, Elmira Daskalova, N. Zheleva, Yana Topalova, Ventseslav Atanasov, Wolfgang Voelter, Pavlina Dolashka

**Affiliations:** 1Institute of Organic Chemistry with Centre of Phytochemistry, Bulgarian Academy of Sciences, Acad. G. Bonchev str., bl.9, 1113 Sofia, Bulgaria; adolashki@yahoo.com (A.D.); ventseslav.atanasov@gmail.com (V.A.); 2Department of General and Applied, Faculty of Biology, Sofia University, St. Kliment Ohridski, Hydrobiology, 8 Dragan Tzankov Blvd., 1164 Sofia, Bulgaria; baba_emi@abv.bg (E.D.); zhelevan@phys.uni-sofia.bg (N.Z.); yanatop@abv.bg (Y.T.); 3Interfacultary Institute of Biochemistry, University of Tübingen, Hoppe-Seyler-Straße 4, D-72076 Tübingen, Germany; wolfgang.voelter@uni-tuebingen.de

**Keywords:** *Cornu aspersum* mucus, antimicrobial peptides, antibacterial activity

## Abstract

Natural products have long played a major role in medicine and science. The garden snail *Cornu aspersum* is a rich source of biologically active natural substances that might be an important source for new drugs to treat human disease. Based on our previous studies, nine fractions containing compounds with Mw <3 kDa; <10 kDa; <20 kDa; >20 kDa; >30 kDa>50 kDa and between 3 and 5 kDa; 5 and 10 kDa; and 10 and 30 kDa were purified from the mucus of *C. aspersum* and analyzed by tandem mass spectrometry (MALDI-TOF/TOF). Seventeen novel peptides with potential antibacterial activity were identified by de novo MS/MS sequencing using tandem mass spectrometry. The different fractions were tested for antibacterial activity against Gram^─^ (*Pseudomonas aureofaciens* and *Escherichia coli*) and Gram+ (*Brevibacillus laterosporus*) bacterial strains as well the anaerobic bacterium *Clostridium perfringens*. These results revealed that the peptide fractions exhibit a predominant antibacterial activity against *B. laterosporus*; the fraction with Mw 10–30 kDa against *E. coli*; another peptide fraction <20 kDa against *P. aureofaciens*; and the protein fraction >20 kDa against the bacterial strain *C. perfringens*. The discovery of new antimicrobial peptides (AMPs) from natural sources is of great importance for public health due to the AMPs’ effective antimicrobial activities and low resistance rates.

## 1. Introduction

Antimicrobial resistance is a major public health problem, which requires scientists and clinicians to identify new efficient antimicrobial agents [1,2,3]. The World Health Organization (2017) composed a list of antibiotic-resistant bacteria for which the development of novel antimicrobial therapies is highly requested, with the global priority on *Acinetobacter baumannii*, *Pseudomonas aeruginosa*, and various species of the *Enterobacteriaceae* family [4].

Antimicrobial peptides (AMPs) are important components of the innate immune system, providing immediate response to a large set of various pathogens such as bacteria, yeasts, fungi, viruses, and even cancer cells [5,6,7,8,9]. These evolutionarily conserved peptides have been found in virtually all organisms ranging from prokaryotes to humans and display a remarkable structural and functional diversity [6]. Conventional AMPs with molecular masses of 1.5–8 kDa are positively charged, with ~15 to 60 amino acid residues and >30% hydrophobic residues [10,11]. AMPs are commonly classified based on their secondary structure into α-helical, β-sheet, or extended/ random-coil structure, and most AMPs belong to the first two groups [3,11].

Since invertebrates lack the adaptive immune system found in vertebrate species, they are reliant solely upon their innate immune systems to counteract invading pathogens. Considering the extraordinary evolutionary success of this group of organisms, it is evident that invertebrate innate immune mechanisms are extremely effective [11]. This has prompted intense studies of invertebrate species in the last few years. Most of the AMPs found in the hemolymph of invertebrates show activity against a mix of microorganisms including bacteria, viruses, and protozoa [12,13,14,15].

Novel proline-rich antimicrobial peptides with molecular masses between 3000 and 9500 Da from the hemolymph of *Rapana venosa* snails were identified [16]. Some of them showed strong antimicrobial activities against *Staphylococcus aureus* (Gram^+^) and low activity against *Klebsiella pneumoniae* (Gram^─^). Several cysteine-rich peptides belonging to the defensin family also were purified from Mediterranean mussels such as myticin from *Mytilus galloprovincialis* [17,18,19]. They are expressed at high levels in hemocytes and are characterized by multiple disulfide bridges, which ensure a precisely folded stable structure for the compact cationic and amphipatic mature peptides. Furthermore, it has been reported that several peptides from the hemolymph of the garden snails *H. lucorum* and *H. aspersa* exhibit a broad spectrum of antimicrobial activity against *S. aureus, S. epidermidis, E. coli, Helicobacter pylori,* and *Propionibacterium acnes* [20,21].

Moreover, the mucus of land snails is a rich source of peptides and proteins with broad-spectrum antibacterial activity. Results indicated that the mucus fraction with Mw between 30 and 100 kDa from the common brown snail *H. aspersa* has a strong antibacterial effect against several strains of *P. aeruginosa* [14]. The identified proteins in *Cornu aspersum* with masses 37.4 kDa, 18.6 kDa, and 17.5 kDa appear also to have activity against *P. aeruginosa* [22]. Additionally, two proteins isolated from the mucus of the African giant land snail *Achatina fulica* were reported to display a broad spectrum of antibacterial activity against *S. aureus* and different strains of *P. aeruginosa* [23]. Furthermore, a novel cysteine-rich antimicrobial peptide mytimacin-AF with potent antimicrobial activity against Gram^─^ and Gram+ bacteria and the fungal strain *C. albicans* was isolated and purified from the mucus of the snail *A. fulica* [23]. Recently, several peptide fractions with antibacterial activities [24,25] and antioxidant properties [26] were isolated from the mucus of the snail *C. aspersum*. Using mass spectrometry, the primary structures of a series of new antimicrobial peptides were determined.

In the present study, the structure of novel peptides and protein fractions isolated from the mucus of the garden snail *C. aspersum* with antibacterial activity are reported. The antibacterial activity of these fractions was tested against four bacterial strains, including Gram^+^ and Gram^−^ bacteria. Bacterial strains *P. aureofaciens* AP9 and *E. coli* NBIMCC 8785 (Gram^−^), *B. laterosporus* BT-271 (Gram^+^), and positive anaerobic spore-forming rod-shaped bacterium *C. perfringens* were chosen because of their antibiotic resistance.

## 2. Materials and Methods

### 2.1. Mucus Collection and Separation of Different Fractions

The mucus was collected and purified from *C. aspersum* snails grown in Bulgarian farms using patented technology without any snail suffering [24]. After ultrafiltration using different membrane filters (10 and 20 kDa), the crude mucus extract was separated into two fractions: a peptide fraction with Mw below 10 kDa and a fraction containing compounds with Mw above 20 kDa. The peptide fraction with Mw below 10 kDa was additionally separated using Amicon^®^ Ultra-15 centrifugal tube filters with 3 and 5 kDa membranes into three fractions. Finally, the following samples were obtained:Sample 1—fraction with compounds of Mw <3 kDaSample 2—fraction with compounds of Mw 3–5 kDaSample 3—fraction with compounds of Mw 5–10 kDaSample 4—fraction with compounds of Mw <10 kDaSample 5—fraction with compounds of Mw 10–30 kDaSample 6—fraction with compounds of Mw <20 kDaSample 7—fraction with compounds of Mw >20 kDaSample 8—fraction with compounds of Mw >30 kDaSample 9—fraction with compounds of Mw >50 kDa

The following membranes were used for ultrafiltration: discs from ultracel regenerated cellulose from 10 kDa NMW, 50 kDa NMW, 100 kDa NMW (EMD Millipore Corporation, Billerica, MA, USA); 1 kDa polyethersulfone membrane filter (Sartorius Stedim Biotech, Göttingen, Germany); and 20 kDa polyethersulfone (Microdyn Nadir™ from STERLITECH Corporation, Goleta, CA, USA).

The protein concentration in the samples was determined by the Bradford assay [27].

### 2.2. Molecular Mass Analysis and de novo Sequencing of Peptides by Mass Spectrometry

The isolated peptide fraction with Mw <3 kDa (Sample 1) was lyophilized and analyzed by MALDI-TOF-TOF mass spectrometry on an AutoflexTM III. High Performance MALDI-TOF& TOF/TOF System (Bruker Daltonics, Bremen, Germany), which uses a 200 Hz frequency-tripled Nd–YAG laser operating at a wavelength of 355 nm. Analysis was carried out after mixing 2.0 μL of the sample with 2.0 μL of matrix solution (7 mg/mL of *α*-cyano-4-hydroxycinnamic acid (CHCA) in 50% ACN containing 0.1% TFA), but only 1.0 μL of the mixture was spotted on a stainless steel 192-well target plate. The samples were allowed to dry at room temperature before being analyzed. A total of 3500 shots were acquired in the MS mode, and collision energy of 4200 was applied. The mixture of angiotensin I, Glu-1-fibrinopeptide B, ACTH (1–17), and ACTH was used for calibration of the mass spectrometer. The MS/MS spectra were carried out in reflector mode with external calibration using fragments of Glu-fibrino-peptide B. Amino acid sequences of peptides were identified by precursor ion fragmentation using MALDI-MS/MS analysis.

### 2.3. Carbohydrate Test

Nine isolated fractions from the mucus were analyzed by the orcinol-sulphuric test to determine the carbohydrate content. About 2 μL of the purified samples were applied to a thin layer plate and air dried. The plate was sprayed with orcinol/H_2_SO_4_ and heated for 20 min at 100 °C. The orcinol/H_2_SO_4_ solution contained 0.02 g of orcinol, 20% H_2_SO_4_, and H_2_O to a total volume of 10 mL.

### 2.4. SDS-PAGE Electrophoresis

Protein fractions with antibacterial activity were analyzed by SDS-PAGE electrophoresis. DL-dithiothreitol, acrylamide/bis-acrylamide (30% solution), bromophenol blue sodium salt (Sigma-Aldrich, Schnelldorf, Germany), N, N, N′, N′-tetramethylethylenediamine (TEMED), ammonium persulphate (APS) (GE Healthcare, Stockholm, Sweden), and Laemmli sample buffer (2×), for SDS PAGE (SERVA, Heidelberg, Germany), were used for electrophoreses analysis. Because commercial Laemmli buffer does not contain any reduction reagent, 10 mM DTT were added as a reducing sample buffer (concentrations refer to 1× sample buffer). Equal volumes containing approximately 25 μg/lane of the samples dissolved in Laemmli sample buffer and protein standard mixture (Precision Plus Protein™, All Blue, Bio-Rad, Feldkirchen, Germany) were separated by 12.5% SDS-PAGE and visualized by staining with Coomassie Brilliant Blue G-250.

### 2.5. Antimicrobial Assays

#### 2.5.1. Microbial Strains

The Gram^+^ bacterial strains of *C. perfringens* NBIMCC 8615, *B. laterosporus* strain BT-271, and Gram^─^ bacterial strains of *P. aureofaciens* AP9 and *E. coli* NBIMCC 8785 were used in the assays of the antibacterial properties of the peptide fractions. These bacterial strains were chosen as models for pathogenic bacteria from different essential Gram^─^ and Gram^+^ groups. Both strains *P. aureofaciens* AP9 and *B. laterosporus* BT-271 were isolated by Topalova (1989) [28] and were characterized as resistant towards aryl-containing xenobiotics and aryl-containing antibiotics possessing the ability to degrade these compounds. *E. coli* NBIMCC 8785 is representative for bacteria from the *Enterobacteriaceae* family and was obtained from the National Bank of Industrial Microorganisms and Cell Cultures (NBIMCC) of Bulgaria. The strain *C. perfringens* [29] was also obtained from the National Bank of Industrial Microorganisms and Cell Cultures.

#### 2.5.2. Nutrient Media and Culture Conditions

Solid MPA (Meat-Peptone Agar) medium/nutrient agar was used to investigate the antibacterial activity through cultivation assays. A nutrient broth was used to multiply the microorganisms. A nutrient liquid media was used to multiply the microorganisms. After rehydration in saline, the strains were maintained on slop agar in standard tubes.

#### 2.5.3. Studies of Antibacterial Activities

The agar diffusion testing developed in [30] is the official method used in many clinical microbiology laboratories for routine antimicrobial susceptibility testing [31,32,33,34].

In the well-known agar disk-diffusion procedure, agar plates are inoculated with a standardized inoculum of the test microorganism. Then, filter paper discs (about 8 mm in diameter), containing the test compound at a desired concentration, are placed on the agar surface.

Another method is agar well diffusion assay. After culturing, the diameter of the inhibition areas is measured after interaction of the antibacterial agents and test cultures. The concentration of the tested samples is determined by the biuret method [35] using bovine serum albumin as standard. The method is based on measuring the amount of peptide bonds. All antibacterial activities were calculated and compared using the quantitative dimension mm^2^/μL sample/mg protein. This allowed for the results to be compared accurately.

Cultivation methods were applied in mesopeptone (nutrient) agar with inoculations run in different conditions:Inoculation was carried out by mixing the standardized microbial suspension with liquid agar at a temperature below 40 °C. With this approach, microorganisms penetrate deep inside the nutrient agar. This procedure is modeling the case in which bacteria develop deeply in the skin. For inoculation, the standardized microbial suspension (50 µL with a density of 10^9^ cell/mL) was spread over the surface of the nutrient solid agar. The peptide fractions (50 µL) were applied in the preliminary prepared wells with a diameter of 8 mm. As the negative control without antibacterial effect, 50 µL distilled water was applied in the wells.Cultivation was performed at 36 °C for 48–72 h for *P. aureofaciens* AP9 and *B. laterosporus* BT-271. Cultivation for these strains was in aerobic conditions. *C. perfringens* NBIMCC 8615 was cultivated in an anaerobic camera /Merck/ in a thermostat at 36 °C for 72 h. *E. coli* was cultured in a thermostat at 36 °C for 48 h.


### 2.6. Electron Microscopy Assays

The impact of active fractions isolated from mucus of the garden snail *C. aspersum* on the cell structure of bacteria was examined by scanning electron microscopy (SEM). All electron microscopic analysis was accomplished by means of SEM—JSM 5510 and LYRA\TESCAN, located in Sofia University “St. Kliment Ohridski”, Faculty of Chemistry and Pharmacy, Bulgaria. The samples were treated by means of increasing concentration of ethyl alcohol and covered with a gold layer before the observation. The images of the Illustration of antibacterial effect of peptide fraction 5 on the cells of *E. coli* are representative micrographs from at least three independent experiments.

## 3. Results

### 3.1. Purification and Characterization of Different Fractions from Mucus

The purified crude mucus extract was separated into various fractions by ultrafiltration under pressure with membrane filters of different pore sizes of 1, 10, and 20 kDa and centrifugal tube filters with 3 and 5 kDa membranes. As a result, 7 fractions were obtained (Sample 1: Mw <3 kDa; Sample 2: Mw 3–5 kDa; Sample 3: Mw 5–10 kDa; Sample 4: Mw <10 kDa; Sample 5: Mw 10–30 kDa; Sample 6: Mw <20 kDa, and Sample 7: Mw >20 kDa).

#### 3.1.1. Molecular Mass Analysis and *de novo* Sequencing of Peptides by Mass Spectrometry

MS analysis of Sample 1 showed several peptides with Mw <3 kDa (Figure 1A). Matrix-assisted laser desorption/ionization time-of-flight mass spectrometry (MS analysis), sequencing the protonated molecular ions [M + H]^+^, was applied to determine the molecular mass and amino acid sequences of the peptides. The amino acid sequences of peptides with low molecular weight were identified by de novo sequencing experiments (MS/MS analysis) of the protonated molecule ions [M + H]^+^. Following b- and y-ions in MS/MS spectrum of peptides at m/z 1059.71 [M + H]^+^, the amino acid sequence MPDGALLGGGGD was identified (Figure 1B).

Using the same method, the amino acid sequences of 17 novel peptides in Sample 1 with molecular masses between 1000 and 2800 Da were identified (Table 1). The isoelectric points (pI) and grand average of hydropathicity (GRAVY) of the peptides were predicted by the ExPASy MW/pI tool program and ExPASy ProtParam tool (Table 1).

#### 3.1.2. Glycosylation Screening

Glycosylation is one of the most prevalent post-translational modifications of proteins, with a defining impact on their structures and functions. The glycosylation of AMPs can influence their antimicrobial activity, and their ability to affect host immunity, target specificity, and biological stability [36,37,38,39]. The favorable impact of glycosylation on pharmacokinetic properties of the native peptides leads to an increase in their oral absorption and bioavailability [37]. The importance of glycosylation has been well studied among the insect AMPs, such as diptericin and formaecin. In all the above-mentioned peptides, the absence of glycosylation abolishes their antimicrobial activity [38,39,40]. Differently glycosylated peptides exhibit differential effects among each other when tested against several Gram-negative bacterial strains. The change of monosaccharide moiety and/or its anomeric configuration in formaecin I and drosocin resulted in a decrease in the antibacterial activity in comparison to that of the native glycopeptide, but the extent of the decrease in antibacterial activity of the glycosylated drosocin analogs was less [40].

Glycosylation of AMP does not necessarily result in generation of an efficacious peptide and can sometimes lead to a loss of activity or functionality [41]. Therefore, it is important to take into consideration the differential expression of peptides among hosts to avoid undesired types of glycosylation [42]. A screening of the fractions from mucus performed by orcinol/H_2_SO_4_ assay identified several glycosylated fractions. As is shown in Figure 2 all fractions except the control fraction water (position 8) change the color in the orcinol–sulfuric acid test applied to the silica gel plate. The peptide fraction with compounds with Mw 3–10 kDa (Figure 2, position 3), and protein fraction 7 with compounds with Mw above 20 kDa (Figure 2, position 7) show a very intensive brown color, probably as more of the compounds in these fraction are glycosylated.

### 3.2. Antibacterial Activity of Different Fractions from Mucus of the Garden Snail C. aspersum

It is known that the Gly and Pro content in peptides plays a crucial role in the activity against different bacteria. Understanding the mechanism of these new antibacterial compounds from the mucus of *C. aspersum* may contribute to the potential of anti-infection therapeutics. Therefore, the antibacterial activity of Sample 1 (compounds with Mw <3 kDa), Sample 2 (compounds with Mw 3–5 kDa), Sample 3 (compounds with Mw 5–10 kDa), Sample 4 (compounds with Mw <10 kDa), Sample 5 (compounds with Mw 10–30 kDa), Sample 6 (compounds with Mw <20 kDa), and Sample 7 (compounds with Mw >20 kDa), isolated from mucus of the garden snail was analyzed against Gram+ bacterial strains of *C. perfringens* and *B. laterosporus* BT-271 and Gram^─^ bacterial strains of *P. aureofaciens* AP9 and *E. coli* NBIMCC 8785.

Two approaches (agar disk-diffusion method and agar well diffusion assay) were used to study antimicrobial effects at depth and on the surface [34]. The surface application gives information about the antibacterial effect when bacteria are located on the surface of the tissue. Applying agar well diffusion assays give information about antibacterial effects of the AMPs when infection is spread deep within the tissues.

The results of tracing the antibacterial activity of nine fractions isolated from mucus against *C. perfringens* NBIMCC 8615 compared to the antibacterial activity against other test cultures (strains) are shown in Figure 3A,B. No antibacterial activity at surface inoculation of the bacterial material is observed in Figure 3A. Antibacterial activity of all tested peptide and protein fractions is consistent with the anaerobic nature of the bacterium. Even though the cultivation is in aerostatic chambers, under anaerobic conditions, neither of the five fractions isolated from the mucus inhibited bacterial strain growth.

The obtained results reveal that three peptide fractions, those with Mw <3 kDa (Sample 1), Mw 3–5 kDa (Sample 2), and Mw 5–10 kDa (Sample 3), have antibacterial activity against the Gram^+^ bacterial strain *B. laterosporus* BT-271, and the most active one is Sample 3 (Figure 3A), which is one of the two fractions with the highest carbohydrate content (orcinol/H_2_SO_4_ test, Figure 2). The compounds with Mw 10–30 kDa (Sample 5) possess a very high inhibition effect against *E. coli* NBIMCC 8785 (Figure 3A). Samples 6, 7, 8, and 9 were also tested in agar disc diffusion tests and they did not show an inhibitory effect.

Comparative analysis of the antibacterial activity of different samples isolated from mucus of the garden snail *C. aspersum* upon deep inoculation of bacteria reveals that Sample 6 with Mw <20 kDa has highest antibacterial activity against the Gram^─^ bacterial strain of *P. aureofaciens* AP9 (3510.00 mm^2^/µL/mg protein) in comparison to that of the other tested fractions, whereas Sample 4 with Mw <10 kDa shows insignificant antibacterial activity against *B. laterosporus* strain BT-271 (Gram^+^) and Gram^─^ bacterial strains of *P. aureofaciens* AP9 and *E. coli* NBIMCC 8785 (Figure 3B) (for Sample 4 Appendix A). Furthermore, only one tested fraction, with high carbohydrate content (Sample 7 with Mw >20 kDa) (Figure 3B) showed high antibacterial activity against *C. perfringens* NBIMCC 8615. This anticlostidial activity is relatively high (1400.17 mm^2^/mg protein/µMol) in comparison to those of the other tested peptide and protein fractions. These results indicate that upon deep inoculation of the microbial material, the protein fraction is of interest for therapeutic purposes against clostridia-induced infections.

The large sterile zones formed due to a strong inhibition effect of Fraction 7 against *C. perfringens* NBIMCC 8615 is illustrated with three repetitions in Figure 4A; they are due to the deep inoculation of the bacteria in comparison to that of the control. For comparison, the antibacterial effect of antimicrobial compounds (AMCs) (Sample 5) against *E. coli* NBIMCC 8785 at surface inoculation of the bacteria is illustrated in Figure 4B. It clearly shows that fraction 7 of the AMCs displays an antibacterial effect against *C. perfringens* NBIMCC 8615.

This allows us to speculate that the fractions may become a candidate for medical treatment of anaerobic infections caused by clostridia. Two fractions with antibacterial activity were subject to SDS-PAGE in order to determine the approximate size of the antimicrobial substances. The electrophoresis (Figure 4C) revealed several compounds in region 10–30 kDa (bands at ~12 kDa, 17–20kDa kDa, ~22 kDa, and between 25 and 30 kDa) and proteins in fraction >20 kDa (bands at ~37 kDa, 42 kDa, between 45 and 50 kDa; ~65kDa, 80 and 90 kDa; 150 and 250 kDa).

To shed light on the mechanism of antibacterial action, we studied the effect of the active fraction with Mw 10–30 kDa on the cell structure of the Gram- bacterial strain *E. coli* NBIMCC 8785 by Scanning Electron Microscopy (SEM).

Figure 5 shows the results obtained by SEM from the control with 18 h culture of *E. coli* in nutrient broth (Figure 5A) and the damaged bacterial wall and membranes of cell of 18 h culture of *E. coli* in nutrient broth after 1 h action of peptide fraction 5 with magnification 10,000× (Figure 5B) and 30,000× (Figure 5C). The deformation of the shape of the bacterial cells can be seen clearly. SEM pictures show the formation of blowing at the two ends as well as a craw in the middle of the bacterial cells.

## 4. Discussion

The discovery of new AMCs from natural sources is of great importance for public health, since these molecules are pharmacological candidates due to their effective antimicrobial activity and low resistance rates. In the case of AMPs, the resistance against them is not prevalent, although AMPs have been exposed to microbes for millions of years [43]. Besides, mutations in the microbes during the course of evolution led to the diversification of AMPs [44].

In our previous work we determined the primary structure and antimicrobial activity of nine peptides produced by the mucus of the garden snail *C. aspersum*, in particular against Gram^─^
*P. aureofaciens* AP9 and Gram^+^
*B. laterosporus* BT271 bacteria [25]. *C. aspersum* and *H. aspersa* are two alternative names for the same species of snail. The taxonomically correct name is *C. aspersum*, but the previous name *H. aspersa* is used more widely. We hypothesized that other peptides from the mucus would also be effective against bacterium *C. perfringens* NBIMCC 8615.

In certain primary structures (Table 1), many peptides contain amino acid residues such as glycine (G), proline (P), leucine (L), valine (V), tryptophan (W), aspartic acid (D), phenylalanine (F), and arginine (R), which are typical for peptides with antimicrobial activity. Many AMPs are unstructured in free solution and fold into their final conformation upon partitioning with biological membranes. Generally, these proteins can attain diverse conformations such as α-helices, β-sheets, mixed conformations, and extended structures [3,44,45,46]. Analyses using the ExPASy ProtParam tool indicate that the mucus fractions with Mw <3kDa contain both cationic and anionic AMPs, but are dominated by cationic AMPs. Most of the peptides identified in fractions with Mw <3kDa are characterized by an amphipathic structure and display generally hydrophobic surfaces (Table 1). This fact is considered as a prerequisite for the disruption of biological membranes and direct cell lysis [3,10,45,46]. It is known that cationic AMPs kill microbes via mechanisms that predominantly involve interactions between the peptide’s positively charged residues and anionic components of target cell membranes. These interactions can then lead to a range of effects including membrane permeabilization, depolarization, leakage, or lysis resulting in cell death. Peptides bind to the membrane surfaces with their hydrophobic sides anchored in the hydrophobic lipid core of the bilayer. There are multiple mechanism models to explain the action of these peptides, including the toroidal pore model, the barrel-stave model, the carpet model, and so on [3,47,48,49]. Some of positively charged AMPs may penetrate into the cell to bind intracellular molecules that are crucial to cell living. They can interact with bacterial ribosomal proteins and induce cell death via interacting with intracellular DNAs and RNAs. [48,50,51]. Recent reports have shown that proline-rich AMPs bind to the 70S ribosome as the main target and interfere with the process of protein synthesis [52,53].

Some of the identified peptides in Table 1 have a primary structure similar to glycine-rich linear antimicrobial peptides, such as Ctenidin1-3 with activity against the Gram^─^ bacterium *E. coli*, isolated and characterized from hemocytes of the spider *Cupiennius salei* [54]. In addition, acanthoscurrins isolated from the hemocytes of the spider *Acanthoscurria gomesiana* act against Gram-negative bacteria, *E. coli*, and the yeast *Candida albicans* and are characterized as cationic peptides with high glycine content [55]. Using alignment of amino acid sequences presented in Table 1 with CAMPSing (http://www.campsign.bicnirrh.res.in/blast.php) [56], the peptides number 6 showed ~72% identity with Ctenidin 1 and Ctenidin 3, as well as ~79% identity with two isoforms of AMP acanthoscurrin (see Appendix A). Peptides numbers 9, 10, 13, and 17 have about 71–76% identity with two isoforms of acanthoscurrin (see Appendix A).

Some of the identified peptides shown in Table 1 belong to anionic antimicrobial peptides (AAMPs), which have been increasingly identified in invertebrates, vertebrates, and plants over the last decade. Previous research also identified AAMPs in mucus fractions with Mw below 10 kDa isolated from garden snail *H. aspersa* [24,25]. Usually, AAMPs show antibacterial activity, but a number of them are multifunctional, variously showing antifungal, anticancer, and neuropeptide activity, and they have the potential to become conventional antibiotics [3,10,57,58]. Some of the antimicrobial mechanisms proposed for these peptides include toroidal pore formation and membrane interaction according to the Shai–Huang–Matsazuki carpet model along with pH-dependent amyloidogenesis and membranolysis via tilted peptide formation [3,10,57,58,59]. AAMPs generally adopt amphiphilic structures. For a number of these peptides, post-translational modifications are essential for antimicrobial activity [53,60,61,62]. Membrane interaction appears to be a key to the antimicrobial function of AAMPs. The architecture of AAMPs is very diverse from alpha-helical peptides in some amphibians to cyclic cysteine knot structures observed in some plant proteins, because of which, in some cases, the mechanisms underlying the antimicrobial action of these peptides are not fully clarified [58,59,63,64,65].

The detected peptides (shown in Table 1, numbers 2, 5, 6, 9, 10, 11, 13, 14, 16, and 17) containing high levels of glycine and leucine residues belong to a new class of Gly/Leu-rich antimicrobial peptides. High homology (above 70.0%, identified from alignment with CAMPSing, (see Appendix A) was found between amino acid sequences of peptide numbers 2, 5, 13, 14, 16, (Table 1) and leptoglycin (GLLGGLLGPLLGGGGGGGGGLL, pI 5.52, GRAVY 1.073) [66], which inhibits the growth of Gram^─^
*P. aeruginosa, E. coli,* and *Citrobacter freundii* strains, but it did not show antimicrobial activity against Gram^+^ bacteria. Our previous studies have also shown the presence of peptides with similar amino acid sequences in the extracts of the garden snail [24,25].

Proline-rich AMPs, with a high content of Pro, Gly, and Arg residues, are an important group of AMPs predominantly active against Gram^─^ bacteria [52,67,68,69]. Previous studies have shown [69], if Pro residues are inserted into the sequences of α-helical AMPs, their ability to permeabilize the bacterial cytoplasmic membrane decreases substantially along with the number of Pro residues incorporated, which could explain our results. Proline-rich peptides, previously known to bind to heat shock proteins, have been shown to inhibit protein synthesis [70,71]. Due to our results from the identified sequences, 12 peptides of the identified sequences contain 1 to 3 Pro amino acid residues in the polypeptide chain. In nine peptides, Pro residues are incorporated in the C-terminal region of the polypeptide chain, but only in 5 peptides are Pro residues located in the N-terminal region. Two peptides contain proline residues both in the N-terminal and C-terminal polypeptide chain. Moreover, one proline residue was found in the center of the polypeptide chain for two peptides (numbers 8 and 15). Although Pro-residue is commonly known as an α-helix breaker, proline residues have been found in the alpha-helical regions of many peptides and proteins, as well as AMPs, such as gaegurin, leptoglycin, buforin, brevinin, and others [66,72,73,74].

The alignment in BLAST has shown, that three peptides (numbers 1, 4, and 8) demonstrate high homology with hemocyanins isolated from the snails *H. aspersa, H. pomatia*, and *H. lucorum* (https://blast.ncbi.nlm.nih.gov) (see Appendix A). Peptide number 1 shows 100% identity with a fragment from the hemocyanins’ subunits β-HaH (*H. aspersa*, Sequence ID: AYO86685.1), β-HlH (*H. lucorum* Sequence ID: AEO51766.1), and β-HpH (*H. pomatia,* Sequence ID: AYO86688.1) and 78% identity with Keyhole limpet hemocyanin. Peptide 4 is 100% identical with the sequence from a fragment of subunits from β-HaH (*H. aspersa*, Sequence ID: AYO86685.1), β-HlH (*H. lucorum,* Sequence ID: AEO51766.1), β-HpH (*H. pomatia,* Sequence ID: AYO86688.1), and α_D_-HaH (*H. aspersa*, Sequence ID: AYO86683.1). Peptide number 8 shows 73% identity with β-HaH, β-HlH, and β-HpH. Probably, proteolytic processes may have led to the appearance of these peptides in the mucus. Furthermore, the majority of known AMPs originate from processing of larger inactive proteins, and some studies suggest that biologically active proteins, such as hemocyanins [75] or hemoglobins [76], can be processed to produce bioactive compounds [13]. Some of the identified peptides contain high levels of glycine and leucine residues, as well as up to thee proline residues, which is probably important for the stability of their antimicrobial activity. A comparison of the alignment of amino acid sequence of the peptides from the mucus of the garden snail *H. aspersa* (presented in Table 1) with databases by software CAMPSing revealed high identification (above 70 %) with known AMPs (see Appendix A). This fact confirms their affiliation with the AMPs family.

In the present study the antibacterial activities of novel peptides and protein fractions isolated from the mucus of the garden snail *C. aspersum* against aerobic bacterial strains *P. aureofaciens* AP9 and *E. coli* NBIMCC 8785, *B. laterosporus* BT-271, and positive anaerobic spore-forming rod-shaped bacterium *C. perfringens* by surface and deep inoculation of the bacteria are presented.

Results from comparative analysis of the antibacterial activity of fractions against *C. perfringens* NBIMCC 8615 (Gram^─^) and aerobic tested bacterial strains *P. aureofaciens* AP9, *E. coli* NBIMCC 8785, *B. laterosporus* BT-271 by surface and deep inoculation of the bacteria reveal that a protein fraction with Mw >20 kDa (one of the two fractions with the highest carbohydrate content) is the most effective against the bacterial strain *C. perfringens* at deep anaerobic cultivation.

Differences in the antibacterial activities of the new peptides and protein fractions isolated from the mucus of the garden snail *C. aspersum* were also found between the aerobic bacterial strains *P. aureofaciens* AP9 and *E. coli* NBIMCC 8785 (Gram^─^) and *B. laterosporus* BT-271 (Gram^+^) upon surface and deep inoculation of the bacteria.

The obtained results reveal that peptide fractions (Samples 1, 2, and 3) exhibit a predominant antibacterial activity against the Gram^+^ bacterial strain *B. laterosporus* BT-271, whereas peptide fractions with Mw <20 kDa have significant antibacterial activity against Gram^─^ bacterial strains *P. aureofaciens* AP9 upon deep inoculation of the bacterium. The presented results show that higher carbohydrate content of the peptide fractions (Samples 1, 2, and 3) leads to higher antibacterial activity against *B. laterosporus* BT-271.

We hypothesize the presence of a synergistic effect of peptides in a fraction below 3 kDa and a fraction of 3–10 kDa and in polypeptides with a molecular weight between 10 and 20 kDa (at ~12 kDa, between 17 and 20 kDa), which is due to the strong antibacterial activity against *P. aureofaciens* AP9 in deep inoculation of the bacterium. The fraction with Mw 10–30 kDa exhibits the highest antibacterial activity using surface inoculation of bacterial strain *E. coli* NBIMCC 8785.

From electrophoresis (Figure 4C), it was clear that the compounds active against *P. aureofaciens* AP9 in the *C. aspersum* mucus were those identified at ~12 kDa, between 17 and 20 kDa, ~22 kDa, and between 25 and 30 kDa. Recently, two peptides (one 17.5 kDa and one 18.6 kDa) were identified in the mucus of *C. aspersum* that appear to have activity against *P. aeruginosa* [22].

It is likely that the proteins determined at ~37 kDa, ~ 42 kDa, 45–50 kDa, ~65 kDa, 80–90 kDa, and between 150 and 250 kDa (electrophoresis, Figure 4C) are responsible for the anticlostidial effect. Some of the found proteins may be related to a new protein named aspernin with a molecular weight of 37.4 kDa, a protein with a molecular weight ~50 kDa, determined previously in fractions from the mucus of *C. aspersum* with anti-pseudomonal properties [22], as well as a protein with Mw 50.81 kDa from *A. fulica* mucus [77]. Furthermore, the protein determined to be between 80 and 90 kDa probably corresponds of a protein with MW of 83.67 kDa (achacin) isolated from *A. fulica* mucus, active against *Streptococcus mutans* and *Actinobacillus actinomycetemcomitans* [77]. Our new data are in agreement with the antimicrobial properties of the mucus from *H. aspersa*, and *A. fulica* [15,22,77].

The combination of two vectors of action of protein fractions >20 kDa against clostridial infections, an antibacterial and a regenerative effect, will be the basis for the development of synergistic therapeutic agents.

Our results show that the antibacterial activity of fractions with Mw 10–30 kDa (Fraction 5 with Mw 10–30 kDa) induces serious damage of the bacterial membrane, changing of the shape and activity of the bacteria strain *E. coli* NBIMCC 8785 (Figure 5A–C). These results will be extended by investigations on the mechanism of the antibacterial effect against *C. perfringens*.

## 5. Conclusions

In conclusion, the present work gives an example of isolation and structural identification of new biomolecules from a complex mixture of *H. aspersa* mucus using advanced, sophisticated instrumentation. As shown through different biotests, several of the mucus fractions have considerable antimicrobial activity, which could probably add to the arsenal of antibiotics as candidates with low resistance rates.

## Figures and Tables

**Figure 1 biomedicines-08-00315-f001:**
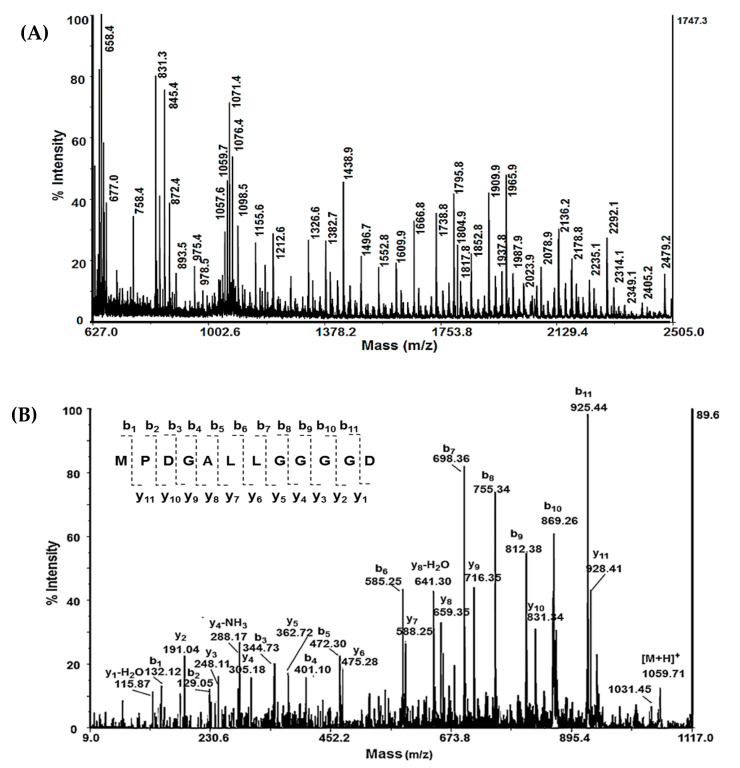
Mass spectrometric analysis of peptide fraction with Mw < 3 kDa by AutoflexTM III, High Performance MALDI-TOF&TOF/TOF Systems (Bruker Daltonics, Bremen, Germany): (**A**) MALDI- MS spectrum; (**B**) MALDI-MS/MS spectrum of peptide [M+H]^+^ at m/z 1059.71 Da.

**Figure 2 biomedicines-08-00315-f002:**
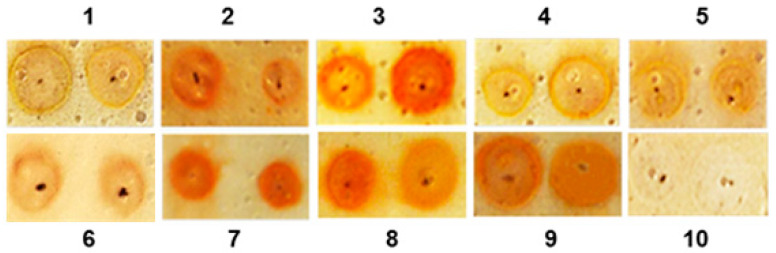
Orcinol–sulphuric acid test applied on to a silica-gel plate of different fractions isolated from the mucus of the garden snail *C. aspersum*. Spots are found in the following positions: position (1), fraction with Mw < 1 kDa; position (2), fraction with Mw < 3 kDa; position (3), fraction with Mw 3–10 kDa; position (4), fraction with Mw 5–10 kDa; position (5), fraction with Mw < 10 kDa; position (6), fraction with Mw < 20 kDa; position (7), fraction with Mw above 20 kDa; position (8), fraction with compounds of Mw > 30 kDa; position (9), fraction with compounds >50 kDa; and position (10), control, containing only water.

**Figure 3 biomedicines-08-00315-f003:**
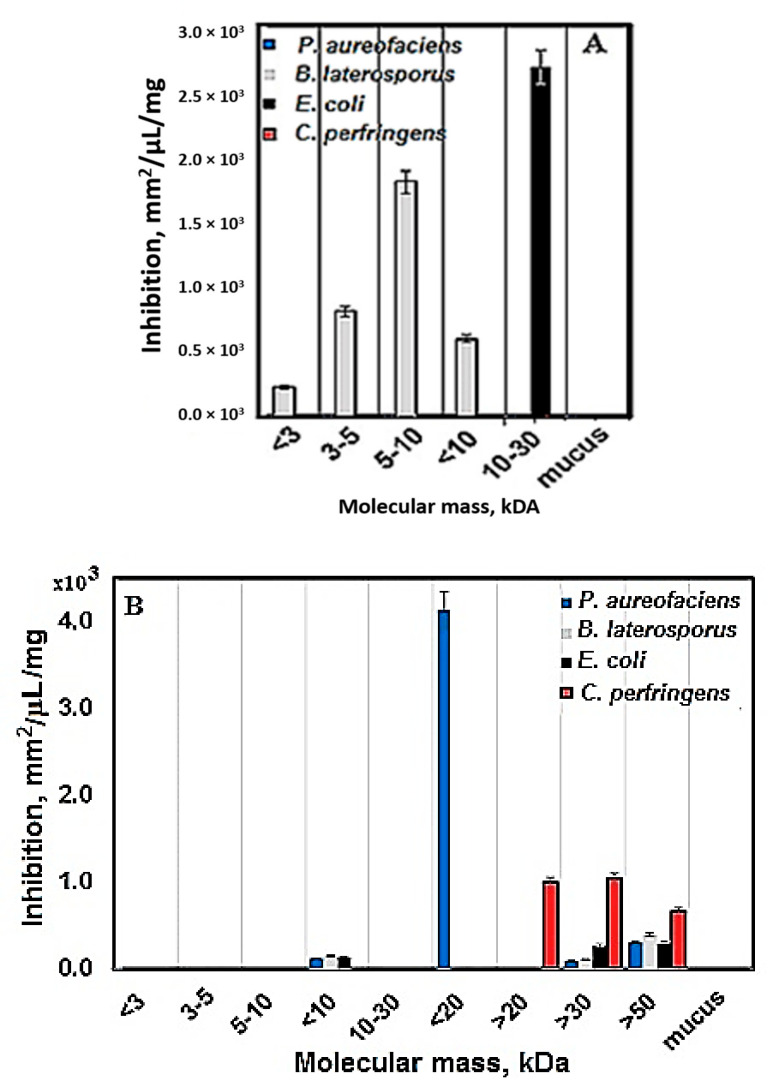
Comparative analysis of the antibacterial activity of Sample 1 (compounds with Mw <3 kDa), Sample 2 (compounds with Mw 3–5 kDa), Sample 3 (compounds with Mw 5–10 kDa), Sample 4 (compounds with Mw <10 kDa), Sample 5 (compounds with Mw 10–30 kDa), Sample 6 (compounds with Mw <20 kDa), Sample 7 (compounds with Mw >20 kDa), Sample 8 (compounds with Mw >30 kDa), Sample 9 (compounds with Mw >50 kDa), and initial mucus extract (mucus) isolated from the garden snail against Gram^+^ bacteria *C. perfringens* and *B. laterosporus* and Gram^_^ bacteria *P. aureofaciens* and *E. coli*: (**A**) surface inoculation of bacteria by agar disk-diffusion assay; (**B**) deep inoculation of bacteria by agar well diffusion assay. Antibacterial activity was measured in inhibition (mm^2^/µL/mg Pr).

**Figure 4 biomedicines-08-00315-f004:**
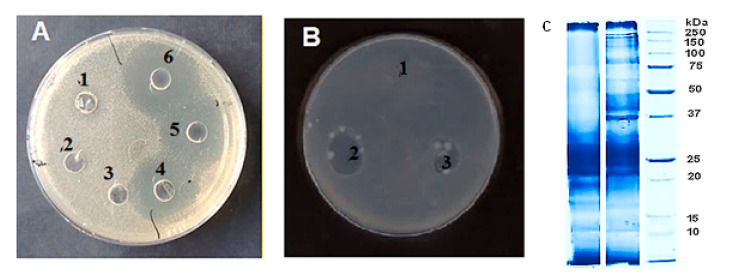
(**A**) Illustrations of sterile areas in investigation of antibacterial activity against the model bacteria of Sample 7 (compounds with Mw >20 kDa), against *C. perfingens* at deep anaerobic cultivation; positions 1, 2, and 3: control; positions 4, 5, and 6: Sample 7. (**B**) Antibacterial activity of Sample 5 (fraction with Mw 10–30 kDa) against *E. coli* NBIMCC 8785 at surface cultivation: position 1: Control, position 2 and 3: Sample 5. (**C**) 12.5% SDS–PAGE with Coomassie Brilliant Blue G-250 staining of protein fractions: position 1 (Samples 5 with Mw 10–30 kDa), position 2 (Samples 7 with Mw >20 kDa), and position 3 (standard protein marker, Protein Prestained Standards, Biorad).

**Figure 5 biomedicines-08-00315-f005:**
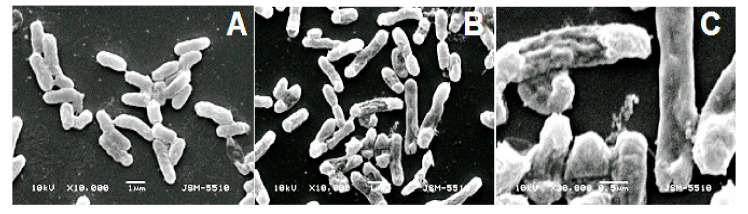
Illustration of antibacterial effect of peptide fraction 5 on the cells of *E. coli*: (**A**) Control, 18 h culture of *E. coli* in nutrient broth; (**B**) Damaged membranes and deformation of cell of 18 h culture of E. *coli* in nutrient broth after 1 h action of peptide fraction 5: 10,000×; (**C**) Damaged membranes and deformation of cell of 18 h culture of *E. coli* in nutrient broth after 1 h action of peptide fraction 5, Magnification 30,000×. The images in Figure 5 are representative micrographs from at least three independent experiments.

**Table 1 biomedicines-08-00315-t001:** Amino acid sequences of peptides from mucus of the garden snail *C. aspersum*, identified by *de novo* sequencing on MALDI-MS/MS.

No	Amino Acid Sequence of Peptides	Exper.Mass [M+H]^+^, Da	Calcul. Monois. Mass, Da	pI	Grand Average of Hydropathicity (GRAVY)	Net Charge
1	DLTLNGLSPK	1057.58	1056.58	5.84	−0.300 (hydrophilic)	−1/+1
2	MPDGALLGGGGD	1059.71	1058.47	3.56	+0.058 (hydrophobic)	−2/0
3	DGPADNAQGAVG	1071.44	1070.46	3.56	−0.600 (hydrophilic)	−2/0
4	SLEERDIQS	1076.44	1075.49	4.14	−0.980 (hydrophilic)	−3/+1
5	GGLLAAGAGGGGAAV	1098.53	1097.58	5.52	+1.200 (hydrophobic)	0/0
6	LGLGNGGAGGGLVGG	1155.57	1154.60	5.52	+0.687 (hydrophobic)	0/0
7	LNLGLDAGGGDPGG	1212.57	1211.58	3.56	−0.093 (hydrophilic)	−2/0
8	FNHKSLPKLEN	1326.64	1325.64	8.60	−1.227 (hydrophilic)	−1/+2
9	NLVGGLSGGGRGGAPGG	1382.70	1381.71	9.75	−0.024 (hydrophilic)	0/+1
10	LGGLGGGGAGGGGLVGEPG	1438.86	1437.72	4.00	+0.439 (hydrophobic)	−1/0
11	NLVGGSGGGGRGGANPLG	1496.73	1495.75	9.75	−0.217 (hydrophilic)	0/+1
12	NGPNGGLGGSLVNGDPK	1552.76	1551.76	5.84	−0.735 (hydrophilic)	−1/+1
13	GLLGGGGGAGGGGLVGGLLNG	1609.94	1608.86	5.52	+0.776 (hydrophobic)	0/+1
14	MGGLLGGVNGGGKGGGGPGAP	1666.83	1665.83	8.50	+0.005 (hydrophobic)	0/+1
15	MLLNAKWAPHSTGPPNA	1804.91	1803.91	8.52	−0.400 (hydrophilic)	0/+1
16	LPFLGLVGGLLGGSVGGGGGGGGPAL	2136.20	2135.17	5.52	+1.023 (hydrophobic)	0/0
17	DVESLPVGGLGGGGGGAGGGGLVGGNLGGGAG	2479.20	2478.21	3.67	+0.353 (hydrophobic)	−2/0

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
