# Peer review of "Antimicrobial Activities of Different Fractions from Mucus of the Garden Snail Cornu aspersum"

_biomedicines, 2020, doi:10.3390/biomedicines8090315_

Round 1

Reviewer 1 Report

  1. The title should be more sugestive and detailed. The latin name of the snail is Cornum aspesum, not C. aspersa.
  2. Key words - the provenience and type of bacterial strains is a detail that should not be specified in this section.
  3. Introduction section, line 57, K. pneumonia should be replaced with K. pneumoniae. Also here the name of the genus, Staphylococcus and Klebsiella should initially written in its complete form.
  4. Lines 64-66 (Cornu aspersum and Helix aspersa are two alternative names for the same species of snail. The taxonomically correct name is C. aspersum, but the previous name H. aspersa is used more widely)are not required in the introduction section.
  5. Lines 138-142 - the provenience of the strains is not necessary. 
  6. Line 144-150 - the description of the media and culture conditions is far too detailed
  7. Line 157 - 10shpuld replace 109.
  8. Line 162 - sterile should be replaced with inhibition, here and in all other cases.
  9. Images 4 A and B could be more clear.

Author Response

  1. The title should be more sugestive and detailed. The latin name of the snail is Cornum spesum, not aspersa.

Answer: text is corrected with Cornu aspersum and the title : Antimicrobial activities of different fractions from mucus of garden snail Cornu aspersum

  1. Key words - the provenience and type of bacterial strains is a detail that should not be specified in this section.

Answer: text is corrected and the type of bacterial strains are removed.

  1. Introduction section, line 57, K. pneumonia should be replaced with K. pneumoniae. Also here the name of the genus, Staphylococcus and Klebsiella should initially written in its complete form.

Answer: text is corrected

  1. Lines 64-66 (Cornu aspersum and Helix aspersumare two alternative names for the same species of snail. The taxonomically correct name is C. aspersum, but the previous name H. aspersumis used more widely)are not required in the introduction section.

Answer: text is replaced in Discussion

  1. Lines 138-142 - the provenience of the strains is not necessary. 

Answer: text is removed

  1. Line 144-150 - the description of the media and culture conditions is far too detailed

Answer: the text is removed

  1. Line 157 - 109 should replace 109.

Answer: the text is corrected

  1. Line 162 - sterile should be replaced with inhibition, here and in all other cases.

Answer: the text is corrected

  1. Images 4 A and B could be more clear.

Answer: figures are corrected

Reviewer 2 Report

In the manuscript called “Antimicrobial compounds from the mucus of garden snail Cornu aspersa” the authors identify and characterize some peptides found in the mucus of the garden snail Cornu aspersa. By using membrane filters of different pore sizes, the authors obtain seven peptide fractions including different mass size peptides. The sequence of the shortest peptides is determined by mass spectrometry, and some physicochemical properties of the peptides are included in the manuscript. The seven peptide fractions are subjected to a glycosylation screening showing that two of them (fractions 3 and 7) show glycosylation. All the fractions are also tested against gram-positive and gram-negative bacteria in both, a surface and a deep inoculation culture in mesopeptone agar plates. Especially, fractions 3, 5, 6 and 7 exhibit an inhibitory effect against the microorganisms. The authors perform an SDS-PAGE gel in order to visualise the main peptides found in fractions 5 and 7. Fraction 5 is also used to visualise its effect in Escherichia coli cells by using scanning electron microscopy.

The experimental approach used in this research concurs with the one exhibited in other papers with a similar final aim, that is, the identification of new antimicrobial compounds from alternative sources, specifically mucus of a garden snail in the present manuscript. Besides minor changes to be considered in the whole manuscript and in the different sections (see the list of comments below), my main concern is the approach used in the discussion. The authors use more than a half of the section to discuss the smallest peptide sequences (sample 1) and their similarity with other antimicrobial peptide sequences. This would be interesting if sample 1 was efficient in the inhibition of the bacterial growth, but this does not happen as seen in Figure 3, where only a low inhibition against B.laterosporus BT-271 is observed. In contrast, the samples 5 and 6, showing inhibition against E.coli NBIMCC8785 and P.aureofaciens AP-9, respectively, are shortly discussed.

In my opinion, more experiments with samples 5, 6 and 7 must be required in order to increase the impact of this publication. These experiments could include the purification of the peptides located in these samples with the goal of identifying the peptides responsible for the inhibition. Working with the purified peptides would allow to measure the MIC and make easier the comparison with other already published AMPs.

To be considered in the whole manuscript:

  • The manuscript lacks enough references to support the hypothesis proposed. It would be mandatory to support the discussion with the introduction of more references.
  • The first time that the name of a microorganism appears, it must be complete. After that, the shorter name can be used (e.g. Lane 57: Staphylococcus aureus and Klebsiella pneumoniae).
  • Be consistent with the use of Gram-positive or Gram+ in the whole manuscript.

Comments by section:

Introduction

  • Lane 41: “Antimicrobial peptides” was written in the abstract, but not in the text before using its acronym (AMP).
  • Lane 43: “These naturally occurring evolutionarily conserved peptides”. This sentence is too long. It is better to split it in two sentences or to remove “naturally occurring”.
  • Lane 57: Gram-positive and Gram-negative
  • Lane 81: The authors chose four bacterial strains in the basis of their antibiotic resistance phenotype, but they don´t explain the experiments that they will perform with them (e.g. the authors could add that they will test the activity of the different isolated peptide fractions by using the four bacterial strains).

Material and methods:

2.1. Mucus collection and separation of different fractions

  • Was the mucus diluted in PBS previous separation by using the filters?
  • Were the 10 and 20 KDa membrane filters bought to Millipore?
  • As I understood, sample 4 would contain the same peptides than samples 1, 2 and 3. Are you interested in sample 4 in order to see synergistic effects between the fractions?
  • Why didn´t exist a sample including 10-20 KDa? You could obtain it by filtering the 20 KDa filtered sample by using a 10 KDa membrane filter.

2.3. Carbohydrate test

  • Did you measure the protein concentration in the samples in order to normalise the amount of peptide included in the test? If you did it, please include in the material and methods section the employed protocol/kit for protein concentration measurement.

2.4. SDS-PAGE electrophoresis

  • Did you perform any two-dimensional (2D-PAGE) gel electrophoresis? I cannot find the figure in the manuscript.
  • In the text is not clear if the authors prepared the gel by using the mentioned reagents (e.g. acrylamide, TEMED, APS…) or they used the pre-cast SERVAGEL TG PRIME gels.
  • Did the Laemmli buffer contain DTT?
  • Include the concentration of the Tris/HCl buffer in the Laemmli recipe.

2.4.1. Microbial strains

  • Lane 113: could you specify what you are talking about in this sentence? “[...] with specific relations towards antibiotics and xenobiotics as well as specific permeability of their cell wall.
  • The acronym NBIMCC appears earlier than National Bank of Industrial Microorganisms and Cell Cultures. Add also that this collection is located in Bulgaria.

2.4.3. Studies of antibacterial activities, using different methods.

  • Lane 151: I would remove “using different methods”. The authors can explain them in the text.
  • Lanes 154-158: Please, explain clearly the two methods. As you call both of them “inoculation” is difficult to differentiate them in the text.
  • Add the conditions (temperature, time) for the E.coli cultivation?
  • In the plates, are you using discs or wells containing the different samples? It must be clear in the text.
  • Include the concentration of the sample before being loaded on the wells or discs.

2.5. Electron microscopic assays

  • Lane 164: Electron microscopy assays
  • Add information about the used microscope and protocol
  • It seems than only sample 5 was used in this experiment. It would be interesting to explain that fact here.
  • Lane 167: “All data are arithmetic averages of three independent repeats.” I don´t understand the sentence in this context.
  1. Results

3.1. Purification and characterization of different fractions from mucus

  • Lane 171: I am aware that the authors referred to another paper in order to explain the protocol of purification of the mucus (lane 88) but I think that additional information would be useful here. Could the authors explain this purification?
  • Lane 172: “membrane filters of different pore sizes of 1, 10 and 20 KDa). I suspect that the authors did not use the 1KDa membrane filter.

3.1.1. Molecular mass analysis and de novo sequencing of peptides by mass spectrometry

  • Table 1. Include why the prolines are underlined and written in red lettering.
  • Could some of the obtained peptides be a product of the cleavage of higher MW proteins included in the mucus?

3.1.2. A glycosylation screening

  • Lane 196: remove “a”
  • Add one sentence explaining why is important to check glycosylation in these peptides.
  • If fraction 4 includes fraction 3,2 and 1, why there is no reaction with orcinol? Is this fact due to a dilution effect?
  • Did you normalize the concentration in every spot before adding 2 ul of the samples to the TLC plate?
  • Negative control negative: Why is the control water? Was also the mucus diluted with water?
  • Did you use a positive control (e.g. mannose) in the test? Did you apply this sample at different concentrations in order to use it as a standard
  • Lane 200: compounds MW 5-10 KDa instead of MW 3-10 KDa

3.2. Antibacterial activity of different fractions form the mucus of garden snail C.aspersum

  • Add references to the two types of inoculation.
  • The figure 3 is very confusing (e.g. it is difficult to know if the blue, red and green bars belong to sample 3 or 4 in the in-depth inoculation). Due to the lack of inhibition exhibited by some sample-microorganism combinations, I suggest to use a table instead of a bar graph in order to present the obtained data.
  • I don’t understand the Y axis legend. I assume that you divided by the protein amount in order to normalise the results, but why by µMol?
  • How did you calculate the protein concentration?
  • Is relevant the low inhibition showed by sample 1 against B.laterosporus BT-271?
  • Please, explain the Figure 4B. Why “six” is written there?
  • What is the negative control in these plates?
  • Could you add a description of the morphological differences between the control and the treated cells in the SEM pictures? (e.g. gross changes in membrane morphology, including roughness)
  1. Discussion
  • More references supporting the different hypothesis suggested by the authors are needed. Also some of them must be changed (e.g. Lane 310: authors refers to a paper about anionic peptides when they are talking about cationic ones ([30] Harris, F.; Dennison, S.R; Phoenix, D.A. Anionic antimicrobial peptides from eukaryotic organisms. Curr Protein Pept Sci. 2009, 10, 585-606). Also in lane 329, the authors refer a cationic paper when they are talking about anionic AMPs (Brown, K. L., & Hancock, R. E. (2006). Cationic host defense (antimicrobial) peptides. Current Opinion in Immunology, 18(1), 24–30. doi:10.1016/j.coi.2005.11.004).
  • Also concerning the references: it is better to refer the primary sources (e.g. Lane 310, better using [20] Matsuzaki K. Control of cell selectivity of antimicrobial peptides. Biochim. Biophys. Acta, 2009, 1788(8): 1687-1692. [21] Giuliani A.; Pirri, G.; Nicoletto, S.F. Antimicrobial peptides: an overview of a promising class of therapeutics. Cent. Eur. J. Biol., 2007, 2, 1-33. [22] Brogden K.A. Antimicrobial peptides: pore formers or metabolic inhibitors in bacteria? Nat. Rev. Microbiol., 2005, 3, 238-250 than [30] Harris, F.; Dennison, S.R; Phoenix, D.A. Anionic antimicrobial peptides from eukaryotic organisms. Curr Protein Pept Sci. 2009, 10, 585-606).
  • Lane 296: remove “only”
  • Lane 301 Aspartic acids. Remove the “s”.
  • Lane 305: loop is not considered a structural conformation for peptides in the AMP database.  Protein loops are patternless regions which connect two regular secondary structures.
  • Lane 302: “Many AMPs are unstructured in solution and change conformation in contact with biological membranes”. They can also change their conformation by adding salts, different pH, in the presence of membrane mimicking systems (e.g. detergent micelles…)
  • Lane 314: “electrostatic difference”. Rewrite it.
  • Lane 314: Add references to the sentence where the AMPs-ribosomes and AMPs-nucleic acids interactions are mentioned.
  • Could a table showing the amino acid conservation between the found peptides (sample 1) and the ones included in the database be more intuitive than the text? Maybe even some alignments (e.g. by using ClustalW) could help in the discussion regarding the similarities between the peptides included in sample 1 and the ones that exhibited antibacterial activity.
  • Lane 332: Shai-Huang-Matsazuki CARPET model. Include “carpet”.
  • Lane 333: AAMPs generally adopt amphiphilic structures. This fact is not only dependent on the negative charged residues, it depends on the global sequence.
  • Lane 334: Add a reference to justify the essentiality of the post-translational modifications in the antimicrobial activity for some peptides.
  • Lanes 325-338: The paragraph concerning AMMPs is not clear. Is the mechanism for AAMPs activity known (lane 330 “the Shai-Huang-Matsazuki model along with pH-dependent amyloidogenesis and membranolysis via tilted peptide formation”) or it is not (lane 337 “the mechanisms underlying the antimicrobial action of these peptides are unclear or have not been elucidated “).
  • Lanes 325-338: add references.
  • Lane 347. Include references to the proline-rich peptides.
  • Lane 356: The insertion of a proline in the middle of a helix would break it making more difficult the interaction with the membrane.
  • Lane 379:  In my opinion, this conclusion is too ambitious. “The presented results show that the higher carbohydrate content of the peptide fractions (Samples 1, 2, and 3) leads to higher antibacterial activity against B. laterosporus BT-271.” It could be also explained by different concentrations in the samples. Why sample 4 has no effect if it includes samples 1,2 and 3?
  • Lane 294: The existence of a band in this position does not mean that this protein is the responsible for the inhibitory effect observed in the bacterial growth. Higher concentration does not mean higher activity.
  • In my opinion, there is too much speculation in the paragraph concerning which protein is the responsible for the inhibition of the bacterial growth. More experiments would be needed in order to clarify this.
  • Lane 408: How do you know that metabolism has changed? In the micrographs, you can only visualise changes in membrane shape. We don´t know what is happening in the metabolism of the bacteria. Or we do?

Author Response

Open Review 2

Comments and Suggestions for Authors

  1. In the manuscript called “Antimicrobial compounds from the mucus of garden snail Cornu aspersa” the authors identify and characterize some peptides found in the mucus of the garden snail Cornu aspersa. By using membrane filters of different pore sizes, the authors obtain seven peptide fractions including different mass size peptides. The sequence of the shortest peptides is determined by mass spectrometry, and some physicochemical properties of the peptides are included in the manuscript. The seven peptide fractions are subjected to a glycosylation screening showing that two of them (fractions 3 and 7) show glycosylation. All the fractions are also tested against gram-positive and gram-negative bacteria in both, a surface and a deep inoculation culture in mesopeptone agar plates. Especially, fractions 3, 5, 6 and 7 exhibit an inhibitory effect against the microorganisms. The authors perform an SDS-PAGE gel in order to visualise the main peptides found in fractions 5 and 7. Fraction 5 is also used to visualise its effect in Escherichia coli cells by using scanning electron microscopy.
  2. The experimental approach used in this research concurs with the one exhibited in other papers with a similar final aim, that is, the identification of new antimicrobial compounds from alternative sources, specifically mucus of a garden snail in the present manuscript. Besides minor changes to be considered in the whole manuscript and in the different sections (see the list of comments below), my main concern is the approach used in the discussion. The authors use more than a half of the section to discuss the smallest peptide sequences (sample 1) and their similarity with other antimicrobial peptide sequences. This would be interesting if sample 1 was efficient in the inhibition of the bacterial growth, but this does not happen as seen in Figure 3, where only a low inhibition against B.laterosporus BT-271 is observed. In contrast, the samples 5 and 6, showing inhibition against E.coli NBIMCC8785 and P.aureofaciens AP-9, respectively, are shortly discussed.

Answer: Sample 1 – fraction with compounds of Mw<3 kDa  - We have discussed the smallest peptide sequences (sample 1) because the same peptides are included in Sample 6 (Sample 6 – fraction with compounds of Mw<20 kDa). By mass spectrometry we can identify the only fraction with Mw lower than 3 kDa.

Compounds in samples 5 and 6 can not be identified because they have higher molecular masses. We have analysed them by 1D-PAGE.  Sample 5 – fraction with compounds of Mw 10-30 kDa.

  1. In my opinion, more experiments with samples 5, 6 and 7 must be required in order to increase the impact of this publication. These experiments could include the purification of the peptides located in these samples with the goal of identifying the peptides responsible for the inhibition. Working with the purified peptides would allow to measure the MIC and make easier the comparison with other already published AMPs.

Answer : If we purify the peptides located in these samples to identify the peptides responsible for the inhibition, we will not get any results, as they will not be active after purification with HPLC system and acetonitrile. To identify the active mucus fractions, we separated them using different membranes and therefore analyzed several samples that contained components with Mw <3 kDa, <20 kDa.

It is important to note that our hypothesis is an inhibitory effect of a complex of several prolines, glycine peptides and some proteins.

  1. To be considered in the whole manuscript:
  • The manuscript lacks enough references to support the hypothesis proposed. It would be mandatory to support the discussion with the introduction of more references.

Answer: To support the discussion several new references are included.

  • The first time that the name of a microorganism appears, it must be complete. After that, the shorter name can be used (e.g. Lane 57: Staphylococcus aureus and Klebsiella pneumoniae).

Answer: We have corrected the names of microorganisms.

  • Be consistent with the use of Gram-positive or Gram+ in the whole manuscript.

Answer: We decided to use “Gram +and  Gram”   throughout the manuscript.

  1. Comments by section:

Introduction

  • Lane 41: “Antimicrobial peptides” was written in the abstract, but not in the text before using its acronym (AMP).

Answer: We fully agree with your proposal and have made an adjustment.

  • Lane 43: “These naturally occurring evolutionarily conserved peptides”. This sentence is too long. It is better to split it in two sentences or to remove “naturally occurring”.

AnswerThank you. We fully agree with your proposal and remove “naturally occurring” in the sentence.

  • Lane 57: Gram-positive and Gram-negative

Answer:  As already mentioned above, we decided to use “Gram +and Gram”   throughout the manuscript.

  • Lane 81: The authors chose four bacterial strains in the basis of their antibiotic resistance phenotype, but they don´t explain the experiments that they will perform with them (e.g. the authors could add that they will test the activity of the different isolated peptide fractionsby using the four bacterial strains).

Answer: We fully agree with your suggestion and have added a new sentence.

“In the present study the structure of novel peptides and protein fractions isolated from the mucus of garden snail C. aspersum with antibacterial activity are reported. The antibacterial activity of these fractions was tested against four bacterial strains, including Gram+ and Gram bacteria.  Bacterial strains P. aureofaciens AP9 and E. coli NBIMCC 8785 (Gram), B. laterosporus BT-271 (Gram+) and positive anaerobic spore-forming rod-shaped bacterium C. perfringens were chosen because of their antibiotic resistance.

  1. Material and methods:

2.1. Mucus collection and separation of different fractions

  • Was the mucus diluted in PBS previous separation by using the filters?

Answer:  Yes, 1:1 (mucus/ PBS)

  • Were the 10 and 20 KDa membrane filters bought to Millipore?

Answer: We used membranes for ultrafiltration as follows: discs from ultracel regenerated cellulose from 10 kDa NMW, 50 kDa NMW, 100 kDa NMW (EMD Millipore Corporation, Billerica, U.S.A.); 1kDa polyethersulfone membrane filter (Sartorius Stedim Biotech, Göttingen, Germany) and 20 kDa polyethersulfone (Microdyn Nadir™ from STERLITECH Corporation U.S.A.).

  • As I understood, sample 4 would contain the same peptides than samples 1, 2 and 3.
  • Are you interested in sample 4 in order to see synergistic effects between the fractions?

Answer:  Exactly the fraction below 10 kDa includes all three low molecular weight fractions obtained after separation with 3 and 5 kDa membranes. Our attention to study all 4 fractions was to check the possibility of a synergistic effect. We observed an effect of fraction <10 kDa with surface inoculation of BT271 (Figure 4 A) and a weak synergistic effect of AP9, BT271 and E. coli in deep inoculation of bacteria (Figure 4 B)

Why didn´t exist a sample including 10-20 KDa? You could obtain it by filtering the 20 KDa filtered sample by using a 10 KDa membrane filter.

Answer: Yes, but we cannot identify the active fractions. Therefore, we focused our study on the fraction with Mw <10 kDa and identified their sequence of amino acids and Mw.

2.3. Carbohydrate test

  • Did you measure the protein concentration in the samples in order to normalise the amount of peptide included in the test? If you did it, please include in the material and methods section the employed protocol/kit for protein concentration measurement.

Answer: The protein concentration in the samples was determined by the Bradford assay [27]

2.4. SDS-PAGE electrophoresis

  • Did you perform any two-dimensional (2D-PAGE) gel electrophoresis? I cannot find the figure in the manuscript.
  • In the text is not clear if the authors prepared the gel by using the mentioned reagents (e.g. acrylamide, TEMED, APS…) or they used the pre-cast SERVAGEL TG PRIME gels.

Answer: Your questions are due to an error (inaccuracy) by us. The all text is corrected as follows:

“Protein fractions with antibacterial activity were analyzed by SDS-PAGE electrophoresis. DL-dithiothreitol, acrylamide/bis-acrylamide (30% solution), bromophenol blue sodium salt (Sigma-Aldrich, Germany), N,N,N',N'-tetramethylethylenediamine (TEMED) and ammonium persulphate (APS) (GE Healthcare, Sweden) and Laemmly sample buffer (2x), for SDS PAGE (SERVA, Germany) were used for electrophoreses analysis.  Because commercial Laemmly buffer does not contain any reduction reagent,  10 mM DTT were added  and received a reducing sample buffer (concentrations refer to 1x sample buffer). Equal volumes containing approximately 25 μg/lane of the samples dissolved in Laemmli sample buffer and protein standard mixture (Precision Plus Protein™, All Blue, Bio-Rad) were separated by 12.5% SDS-PAGE and visualized by staining with Coomassie Brilliant Blue G-250. “

  • Did the Laemmli buffer contain DTT? •
  • Include the concentration of the Tris/HCl buffer in the Laemmli recipe.

Answer: We used commercial Laemmli sample buffer (2x), for SDS PAGE (SERVA, Germany), that does not contain any reduction reagent.  Before use we added 10 mM DTT and receive a reducing sample buffer (concentrations refer to 1x sample buffer). We prepared the separating gel and stacking gel for SDS-PAGE electrophoresis and did not used the pre-cast SERVAGEL TG PRIME gels.

2.4.1. Microbial strains

  • Lane 113: could you specify what you are talking about in this sentence? “[...] with specific relations towards antibiotics and xenobiotics as well as specific permeability of their cell wall.

Answer : This sentence was removed

  • than National Bank of Industrial Microorganisms and Cell Cultures. Add also that this collection is located in Bulgaria.

Answer: We added it.

2.4.3. Studies of antibacterial activities, using different methods.

  • Lane 151: I would remove “using different methods”.

Answer: Thanks for your suggestion. We have removed “using different methods" in the text.

Lanes 154-158: Please, explain clearly the two methods. As you call both of them “inoculation” is difficult to differentiate them in the text.

Answer : Line 157–164 This is a method in which different ways of inoculating the microbial material are used. In the first case, the microbial material is resuspended in molten and cooled to 43 ° C nutrient agar. In the other case, the same amount of microbial suspension is distributed evenly over the surface of the pre-spilled and hardened nutrient agar.

  • Add the conditions (temperature, time) for the coli cultivation?

Answer: Line 180   E. coli was cultured in a thermostat at 36°C for 48 hours.

  • Include the concentration of the sample before being loaded on the wells or discs.

Answer: Line 164-167 The concentration of the tested samples is determined by the biuret method for the determination of protein [34] using bovine serum albumin as standard. The method is based on measuring the amount of peptide bonds. All antibacterial activities were calculated and compared using the quantitative dimension mm2/μl sample/mg protein. This allows the results to be compared accurately.

2.5. Electron microscopic assays

  • Lane 164: Electron microscopy assays

Answer: The correction has been made. Thank you.

  • Add information about the used microscope and protocol

Answer: All electron microscopic analysis were accomplished by means of SEM - JSM 5510 и LYRA\TESCAN, located in Sofia University “St. Kliment Ohridski” – Faculty of Chemistry and Pharmacy, Bulgaria. The samples were treated by means of increasing concentration of ethil alcochol and covered with gold layer before the observation.

The text was added in the draft of paper.

  • It seems than only sample 5 was used in this experiment. It would be interesting to explain that fact here.

Answer: The results for the all investigated protein fraction are presented in quantitative way in mm2/µL/mg protein. Illustration was added only for the sample 5.

  • Lane 167: “All data are arithmetic averages of three independent repeats.” I don´t understand the sentence in this context.

Answer: “All data are average values of the data of three independent repeats.” Every protein fraction was investigated in three repetitions. From three obtained data the average value was calculated. The expression was corrected in the text.

  1. Results

3.1. Purification and characterization of different fractions from mucus

  • Lane 171: I am aware that the authors referred to another paper in order to explain the protocol of purification of the mucus (lane 88) but I think that additional information would be useful here. Could the authors explain this purification?

Answer: The method of extraction and purification of the mucus is subject to patent protection (Dolashka P., BG Useful model 2097/2015). In general, it is used a special device in which the snails are placed with a small amount of distillated water and electrical stimulation with low voltage is applied. It is enough for stimulate snails to release mucus and to keep them alive. Using this method the snails remain alive without disturbing their biological functions and return to the farm. The method can be used repeatedly to extract mucus. Thus obtained crude mucus extract was homogenized and subjected to centrifugation to remove coarse impurities. The supernatant was subjected to several cycles of filtration, using filters with smaller pore sizes for each subsequent filtration.

  • Lane 172: “membrane filters of different pore sizes of 1, 10 and 20 KDa). I suspect that the authors did not use the 1KDa membrane filter.

Answer:  We confirm that we have used just such a filter to remove secondary metabolites with low molecular masses from peptides. We used a Polyethersulfone membrane filter (1kDa) from Sartorius Stedim Biotech (Order No 14609—47------D; Lot No 0709146090260803).

3.1.1. Molecular mass analysis and de novo sequencing of peptides by mass spectrometry

  • Table 1. Include why the prolines are underlined and written in red lettering.

Answer:  Proline-residues are important for antimicrobial activity.

  • Could some of the obtained peptides be a product of the cleavage of higher MW proteins included in the mucus?

Answer: Is it possible some of identified peptifes to be a part of proteins after proteolysis. In the text it presented several peptides that demonstrate high homology with hemocyanins isolated from snails Helix aspersa, Helix pomatia, and Helix lucorum. Probably proteolytic processes may have led to the appearance of these peptides in the mucus. It is known that in the H. aspersa mucus contain small quantity hemocyanin.

3.1.2. A glycosylation screening

  • Lane 196: remove “a”
  • Add one sentence explaining why is important to check glycosylation in these peptides.

Answer: We have removed “a" in the text and explained the importance of the glycosylation of antimicrobial peptides.

“Glycosylation is one of the most prevalent post-translational modifications of proteins, with a defining impact on their structures and functions. The glycosylation of AMPs can influence their antimicrobial activity, and their ability to affect host immunity, target specificity, and biological stability [36-39]. The favourable impact of glycosylation on pharmacokinetic properties of the native peptides leads to an increase in their oral absorption and bioavailability [37]. The importance of glycosylation has been well studied among the insect AMPs, such as diptericin and formaecin and the bacteriocin-family member, enterocin F4-9 In all the above-mentioned peptides, the absence of glycosylation abolishes their antimicrobial activity [38-40]. Differently glycosylated peptides exhibit differential effect among each other when tested against several Gram-negative bacterial strains. The change of monosaccharide moiety and/or its anomeric configuration in formaecin I and drosocin resulted into decrease in the antibacterial activity in comparison to that of the native glycopeptide, but the extent of decrease in antibacterial activity of glycosylated drosocin analogs was less [40]. Glycosylation of AMP does not necessarily result in generation of an efficacious peptide and can sometimes lead to a loss of activity or functionality [41]. Therefore, it is important to take into consideration the differential expression of peptides among hosts to avoid undesired types of glycosylation [42].”

  • If fraction 4 includes fraction 3,2 and 1, why there is no reaction with orcinol? Is this fact due to a dilution effect?

Answer: In fact, fraction 4 reacted positively to the orcinol / H2SO4 test, but the staining was less intense than fraction with Mw < 3 kDа (position2) and  fraction with Mw 3-10 kDа (position 3).

  • Did you normalize the concentration in every spot before adding 2 ul of the samples to the TLC plate?

Answer: The concentration of the samples before to be applied in every spot to the TLC plate was determined because the same samples were used for antimicrobial analysis.

  • Negative control negative: Why is the control water? Was also the mucus diluted with water?

Answer: For orcinol test the mucus was diluted with destilate water which does not give a positive reaction with orcinol. Therefore, the negative control is water.

  • Did you use a positive control (e.g. mannose) in the test? Did you apply this sample at different concentrations in order to use it as a standard

Answer: The mannose test as a positive control is used separately to check the reagents. We apply this sample in different concentrations to use it as a standard, but when testing the individual fractions, the aim is to determine whether there are glycans or not, not the concentration. The examination of the carbohydrate structure of the fraction, which showed activity against bacteria, will be analyzed by mass spectrometric analyzes.

Lane 200: compounds MW 5-10 KDa instead of MW 3-10 Kda

Answer: The text is corrected.

3.2. Antibacterial activity of different fractions form the mucus of garden snail C.aspersum

Add references to the two types of inoculation.

Answer: A new reference was added [34]:

  • The figure 3 is very confusing (e.g. it is difficult to know if the blue, red and green bars belong to sample 3 or 4 in the in-depth inoculation). Due to the lack of inhibition exhibited by some sample-microorganism combinations, I suggest to use a table instead of a bar graph in order to present the obtained data.

Answer: The figure 3 is corrected with the same colure.

  • I don’t understand the Y axis legend. I assume that you divided by the protein amount in order to normalise the results, but why by µMol?

Answer: The dimencion of antibacterial effect is mm2/mkL/mg pr. or mm/µL/mg protein. The mistake was corrected. Thank you for the remark.

  • How did you calculate the protein concentration?

Answer: Protein content in samples has been determined by the method of Kochetov (1974), using bovine serum albumin as standard. This method is based on the measuring of the quantity of peptide bonds.

  • Please, explain the Figure 4B. Why “six” is written there?

Answer : The figure was replaced.

  • What is the negative control in these plates?

Answer: The negative control in the plates was application in the wells 50 µL destilled water.

  • Could you add a description of the morphological differences between the control and the treated cells in the SEM pictures? (e.g. gross changes in membrane morphology, including roughness)

Answer: The explanation text was added in the SEM pictures.

  1. Discussion
  • More references supporting the different hypothesis suggested by the authors are needed. Also some of them must be changed (e.g. Lane 310: authors refers to a paper about anionic peptides when they are talking about cationic ones ([30] Harris, F.; Dennison, S.R; Phoenix, D.A. Anionic antimicrobial peptides from eukaryotic organisms. Curr Protein Pept Sci. 2009, 10, 585-606). Also in lane 329, the authors refer a cationic paper when they are talking about anionic AMPs (Brown, K. L., & Hancock, R. E. (2006). Cationic host defense (antimicrobial) peptides. Current Opinion in Immunology, 18(1), 24–30. doi:10.1016/j.coi.2005.11.004).

Answer: Thank you, we have made the necessary references adjustment

  • Also concerning the references: it is better to refer the primary sources (e.g. Lane 310, better using [20] Matsuzaki K. Control of cell selectivity of antimicrobial peptides. Biochim. Biophys. Acta, 2009, 1788(8): 1687-1692. [21] Giuliani A.; Pirri, G.; Nicoletto, S.F. Antimicrobial peptides: an overview of a promising class of therapeutics. Cent. Eur. J. Biol., 2007, 2, 1-33. [22] Brogden K.A. Antimicrobial peptides: pore formers or metabolic inhibitors in bacteria? Nat. Rev. Microbiol., 2005, 3, 238-250 than [30] Harris, F.; Dennison, S.R; Phoenix, D.A. Anionic antimicrobial peptides from eukaryotic: Curr Protein Pept Sci. 2009, 10, 585-606).

Answer: Thank you. We have made the necessary references adjustment.

  • Lane 296: remove “only”
  • Lane 301 Aspartic acids. Remove the “s”.

Answer: Thank you. The corrections have been made in the text.

Lane 305: loop is not considered a structural conformation for peptides in the AMP database.  Protein loops are patternless regions which connect two regular secondary structures.

Answer: The sentence has been corrected and “loops” was removed.

  • Lane 302: “Many AMPs are unstructured in solution and change conformation in contact with biological membranes”. They can also change their conformation by adding salts, different pH, in the presence of membrane mimicking systems (e.g. detergent micelles…)

Answer: Thank you. You are absolutely right. But in this case we comment on what changes the peptides undergo when in contact with bacterial membranes to show their antibacterial activity

  • Lane 314: “electrostatic difference”. Rewrite it.
  • Lane 314: Add references to the sentence where the AMPs-ribosomes and AMPs-nucleic acids interactions are mentioned.

Answer: We have rewritten this sentence and added the necessary references. We changed the sentence as follows:

“Generally, the cationic AMPs interact with negatively charged bacterial cell membranes through electrostatic interactions and undergo membrane adsorption and conformational change. Peptides bind to the membrane surfaces with their hydrophobic sides anchored in the hydrophobic lipid core of the bilayer. There are multiple mechanism models to explain the action of these peptides, including the toroidal pore model, the barrel-stave model, and the carpet model and so on [3,47-49]. Some of positively charged AMPs may penetrate into the cell to bind intracellular molecules which are crucial to cell living. They can interact with bacterial ribosomal proteins, and induce cell death via interacting with intracellular DNAs and RNAs. [48,50,51]. Recent reports have shown that proline-reach AMPs bind to the 70S ribosome as the main target and interfere with the process of protein synthesis [52,53].”

  • Could a table showing the amino acid conservation between the found peptides (sample 1) and the ones included in the database be more intuitive than the text? Maybe even some alignments (e.g. by using ClustalW) could help in the discussion regarding the similarities between the peptides included in sample 1 and the ones that exhibited antibacterial activity.

Answer:  This information is included in Supplement information 1. The similarity of some identified peptides from C. aspersum mucus with known antimicrobial peptides was detected by alignment of amino acid sequences of these peptides with data base AMPs by CAMPSing (http://www.campsign.bicnirrh.res.in/blast.php) and with proteins by BLAST. Results are presented in Supplement Information 1.

Lane 332: Shai-Huang-Matsazuki CARPET model. Include “carpet”.

Answer: Thank you for the suggestion, we included “carpet”.

  • Lane 333: AAMPs generally adopt amphiphilic structures. This fact is not only dependent on the negative charged residues, it depends on the global sequence.

Answer: We agree and made a correction in the sentence. “AAMPs generally adopt amphiphilic structures depending on the overall sequence and negative charged residues”

  • Lane 334: Add a reference to justify the essentiality of the post-translational modifications in the antimicrobial activity for some peptides.

Answer: Thanks for the suggestion, we included additional references.

  • Lanes 325-338: The paragraph concerning AMMPs is not clear. Is the mechanism for AAMPs activity known (lane 330 “the Shai-Huang-Matsazuki model along with pH-dependent amyloidogenesis and membranolysis via tilted peptide formation”) or it is not (lane 337 “the mechanisms underlying the antimicrobial action of these peptides are unclear or have not been elucidated “).

Answer: We changed the paragraph as follows:

“Some of antimicrobial mechanisms proposed for these peptides include toroidal pore formation and membrane interaction according to the Shai-Huang-Matsazuki carpet model along with pH-dependent amyloidogenesis and membranolysis via tilted peptide formation [3,10,57-59]. AAMPs generally adopt amphiphilic structures. For a number of these peptides, post-translational modifications are essential for antimicrobial activity [53,60-62]. Membrane interaction appears a key to the antimicrobial function of AAMPs. The architectures of AAMPs is very diverse from alpha-helical peptides in some amphibians to cyclic cysteine knot structures observed in some plant proteins, because of which in some cases, the mechanisms underlying the antimicrobial action of these peptides are not fully clarified [58,59,63-65].”

  • Lanes 325-338: add references.
  • Lane 347. Include references to the proline-rich peptides.

Answer: Thanks for the suggestion, we included additional references.

“Proline-rich AMPs, with a high content of Pro, Gly and Arg residues, are an important group of AMPs predominantly active against Gram- bacteria [52,67-69]. Previous studies have shown [69], if Pro residues are inserted into the sequences of α-helical AMPs, their ability to permeabilize the bacterial cytoplasmic membrane decreases substantially along with the number of Pro residues incorporated, what could explain our results. Proline-rich peptides, previously known to bind to heat shock proteins, are shown to inhibit protein synthesis [70,71].”

  • Lane 356: The insertion of a proline in the middle of a helix would break it making more difficult the interaction with the membrane. Поставянето на пролин в средата на спирала би го нарушило, което би затруднило взаимодействието с мембраната.

Answer: Although Pro-residue is commonly known as a helix breaker, proline residues have been found in the alpha-helical regions of many peptides and proteins. The antimicrobial peptide gaegurin displays alpha-helical structure and has a central proline residue [Suh, et al., 1999]. However, peptide Leptoglycin (GLLGGLLGPLLGGGGGGGGGLL) from the South American frog Leptodactylus pentadactylus is active agaist Gram– bacteria (Pseudomonas aeruginosa, Escherichia coli, and Citrobacter freundii) although included Pro-residue in the midel of its polypeptide chain. Other AMPs with Pro-residue are buforin and brevinin from amphibian species, cecropin from the cecropia moth.

  • Lane 379:  In my opinion, this conclusion is too ambitious. “The presented results show that the higher carbohydrate content of the peptide fractions (Samples 1, 2, and 3) leads to higher antibacterial activity against laterosporus BT-271.” It could be also explained by different concentrations in the samples. Why sample 4 has no effect if it includes samples 1,2 and 3?

Answer: Fig. 3A confim the effect of compounds with Mw <10 kDa and our suggestion is based on the ather results published in the journals for glycopeptides.

  • Lane 294: The existence of a band in this position does not mean that this protein is the responsible for the inhibitory effect observed in the bacterial growth. Higher concentration does not mean higher activity.

Answer: Yes, we are agreed. This is only our suggestion that this protein is the responsible for the inhibitory effect observed in the bacterial growth.

  • In my opinion, there is too much speculation in the paragraph concerning which protein is the responsible for the inhibition of the bacterial growth. More experiments would be needed in order to clarify this.

Lane 408: How do you know that metabolism has changed? In the micrographs, you can only visualise changes in membrane shape. We don´t know what is happening in the metabolism of the bacteria. Or we do?

Answer: This is just our assumption that the metabolism has changed. We plan to analyze the protein expression of 2D-PAGE and will publish the results in the next publication to explain the mechanism. 

Round 2

Reviewer 2 Report

In general, the authors have improved the contents of the manuscript (including the addition of samples 8 and 9), but some minor corrections must be done before publication:

1. Correct “Laemmly buffer” by ”Laemmli buffer”

2. In the section “2.4.1. Microbial strains” there is a paragraph in which something is missing: “These bacterial strains were chosen as models for pathogenic bacteria from different essential […] Both strains”

3. In the section “2.5. Electron microscopy assays”: “All data are values of three independent repeats” must be changed to “the images exhibit in figure 5 are representative micrographs from at least three independent experiments”.

4. Table 1: there is a non-underlined Proline (peptide number 1)

5. In the section “3.1.2. Glycosilation screening”: ”Enterocin F4-9” needs a full stop.

6. Were samples 8 and 9 used in the glycosylation screening? Add it to the text.

7. Figure 3: In my opinion, the results can be analysed clearly in this representation but I have some concerns with the Y axis:

- I think that it is better to name it “inhibition” or “inhibitory effects” instead of “effects”

- Correct the magnitude in the Y axis: “2” is a superscript.

- Correct the magnitude in the Y axis:”mkL”” must be “µL”

- Correct the magnitude in the Y axis:”mgPr”” must be “mg” (it is assumed that it is mg of protein)

- Legend of the microorganisms: I would add the name of the microorganisms instead of the strain number. You could use P.aureofaciens, C.perfringens and B.laterosporus.

8. Figure 3: Sample 4 (<10KDa) didn’t exhibit effect against B.laterosporus in the previous version of the manuscript. Why is now different?

9. In section “3.2. Antibacterial activity of different fractions form the mucus of garden snail C.aspersum”:

- “The results of tracing the antibacterial activity of seven fractions isolated from mucus against C.perfringens NBIMCC 8615 compared to the antibacterial activity against other test cultures (strains) are shown in Figures 3A,B.” Change seven to nine fractions.

- Correct “figrue” in: “No surface inoculation of the bacterial material is observed in “figrue 3A”.

- I assume that samples 6, 7, 8 and 9 were also tested in the agar disc diffusion assays. Add to the text that they didn’t exhibit any inhibitory effect. Adding it, the reader can know that they were tested.

- “No surface inoculation of the bacterial material is observed in Figure 3A”. I don´t understand this sentence.

- “AMCs” appears for the first time without adding “antimicrobial compounds”. This was added to the discussion but it was not the first time “AMCs” appears.

- “[…] P. aureofaciens AP9 (3510.00 mm2/mg protein/μMol) in comparison to the other tested fractions […]”. The units are not in concordance with the units written in the graph.

- Some of antimicrobial mechanisms proposed … Add "the".

10. In the Discussion:”the resistance against them is not prevalent although AMPs […]”. I assume that you are talking about AMPs but it is not clear in the sentence. You could use "In the case of AMPs, the resistance against them is not prevalent…"

11. Third paragraph: The mechanism of AMP-membrane interactions is explained twice: “It is known that cationic AMPs kill microbes via mechanisms that predominantly involve interactions between the peptide’s positively charged residues and anionic components of target cell membranes. These interactions can then lead to a range of effects including membrane permeabilization, depolarization, leakage or lysis resulting in cell death.” AND “Generally, the cationic AMPs interact with negatively charged bacterial cell membranes through electrostatic interactions and undergo membrane adsorption and conformational change.”

11. The alignment in BLAST has shown, that three peptides (Nos 1 and 4). Which is the third peptide? Add peptide number 8.

12. Correct “perringens” in “[…](one of the two fractions with the highest carbohydrate content) is the most effective against the bacterial strain C. perringens at deep anaerobic cultivation.”

13. ”From electrophoresis (Fogure 4C)it was CLEAR that […]”.

14. "Our results show that the antibacterial activity of fraction with Mw 10-30 kDa (Fraction 5 with MW 10-30 kDa) induces serious damaging of the bacterial membrane changing of the shape, activity and metabolism of the bacteria strain E. coli NBIMCC 8785 (Figures 5 A, B, C)."

I understand that this research is going to be extended in the future but, at this point you cannot assert that the metabolism of the bacteria is affected only supported by EM images. 

Author Response

I would like to express our great gratitude to the reviewers for the recommendations made and extremely useful corrections to the article presented by us. These adjustments are extremely useful for us.

In general, the authors have improved the contents of the manuscript (including the addition of samples 8 and 9), but some minor corrections must be done before publication:

1.Correct “Laemmly buffer” by ”Laemmli buffer”

Answer : The text is corrected

2.In the section “2.4.1. Microbial strains” there is a paragraph in which something is missing: “These bacterial strains were chosen as models for pathogenic bacteria from different essential […] Both strains”

Answer : The text is corrected

3.In the section “2.5. Electron microscopy assays”: “All data are values of three independent repeats” must be changed to “the images exhibit in figure 5 are representative micrographs from at least three independent experiments”.

Answer : The text is corrected

4.Table 1: there is a non-underlined Proline (peptide number 1)

Answer : Prolin is underlined peptide number 1

5.In the section “3.1.2. Glycosilation screening”: ”Enterocin F4-9” needs a full stop.

Answer : The text is corrected

6.Were samples 8 and 9 used in the glycosylation screening? Add it to the text.

Answer : Figure 2 was corrected and the results for Sample 8 – fraction with compounds of Mw >30 kDa) and Sample 9 – fraction with compounds of  >50 kDa) were added.

7.Figure 3: In my opinion, the results can be analysed clearly in this representation but I have some concerns with the Y axis:

- I think that it is better to name it “inhibition” or “inhibitory effects” instead of “effects”

- Correct the magnitude in the Y axis: “2” is a superscript.

- Correct the magnitude in the Y axis:”mkL”” must be “µL”

- Correct the magnitude in the Y axis:”mgPr”” must be “mg” (it is assumed that it is mg of protein)

- Legend of the microorganisms: I would add the name of the microorganisms instead of the strain number. You could use P.aureofaciens, C.perfringens and B.laterosporus.

Answer : Figure 3 was corrected.

8.Figure 3: Sample 4 (<10KDa) didn’t exhibit effect against B.laterosporusin the previous version of the manuscript. Why is now different?

Answer: In our first version, we did not analyze the surface inoculation by agar disk diffusion test for sample 4. We did additional study of this fraction and added it.

9.In section “3.2. Antibacterial activity of different fractions form the mucus of garden snail C.aspersum”:

- “The results of tracing the antibacterial activity of seven fractions isolated from mucus against C.perfringens NBIMCC 8615 compared to the antibacterial activity

Answer : The text is corrected –“seven fractions” is changed to “nine fractions”

- “The results of tracing the antibacterial activity of seven fractions isolated from mucus against C.perfringens NBIMCC 8615 compared to the antibacterial activity against other test cultures (strains) are shown in Figures 3A,B.” Change seven to nine fractions.

- Correct “figrue” in: “No surface inoculation of the bacterial material is observed in “figrue 3A”.

Answer : The text is corrected

- I assume that samples 6, 7, 8 and 9 were also tested in the agar disc diffusion assays. Add to the text that they didn’t exhibit any inhibitory effect. Adding it, the reader can know that they were tested.

Answer : A new text is added : Samples 6, 7, 8 and 9 were also tested in agar disc diffusion tests and they did not show an inhibitory effect.

- “No surface inoculation of the bacterial material is observed in Figure 3A”. I don´t understand this sentence.

Answer: A new text is added : No antibacterial activity at surface inoculation of the bacterial material is observed in Figure 3A”.

- “AMCs” appears for the first time without adding “antimicrobial compounds”. This was added to the discussion but it was not the first time “AMCs” appears.

Answer: This is corrected. The “antimicrobial compounds (AMCs)” was added on page 8, wehe the first time “AMCs” appears.

- “[…] P. aureofaciens AP9 (3510.00 mm2/mg protein/μMol) in comparison to the other tested fractions […]”. The units are not in concordance with the units written in the graph.

Answer:  text is corrected mm2/µL/mg protein)

- Some of antimicrobial mechanisms proposed … Add "the".

Answer: This is corrected in the text.

  1. In the Discussion:”the resistance against them is not prevalent although AMPs […]”. I assume that you are talking about AMPs but it is not clear in the sentence. You could use "In the case of AMPs, the resistance against them is not prevalent…"

Answer: We agree with your suggestion.  The text is corrected.

  1. Third paragraph: The mechanism of AMP-membrane interactions is explained twice: “It is known that cationic AMPs kill microbes via mechanisms that predominantly involve interactions between the peptide’s positively charged residues and anionic components of target cell membranes. These interactions can then lead to a range of effects including membrane permeabilization, depolarization, leakage or lysis resulting in cell death.” AND “Generally, the cationic AMPs interact with negatively charged bacterial cell membranes through electrostatic interactions and undergo membrane adsorption and conformational change.”

Answer : Thank you. You are absolutely right. The sentence “Generally, the cationic AMPs interact with negatively charged bacterial cell membranes through electrostatic interactions and undergo membrane adsorption and conformational change.” has been removed from the text.

  1. The alignment in BLAST has shown, that three peptides (Nos 1 and 4). Which is the third peptide? Add peptide number 8.

Answer: The peptide number 8 was added.

  1. Correct “perringens” in “[…](one of the two fractions with the highest carbohydrate content) is the most effective against the bacterial strain C. perringens at deep anaerobic cultivation.”

Answer : Thank you. The error has been fixed. The “C. Perringens” was corrected to “C. perfringens”

  1. ”From electrophoresis (Fogure 4C)it was CLEAR that […]”.

Answer : The error has been fixed.

  1. "Our results show that the antibacterial activity of fraction with Mw 10-30 kDa (Fraction 5 with MW 10-30 kDa) induces serious damaging of the bacterial membrane changing of the shape, activity and metabolism of the bacteria strain E. coli NBIMCC 8785 (Figures 5 A, B, C)."

I understand that this research is going to be extended in the future but, at this point you cannot assert that the metabolism of the bacteria is affected only supported by EM images. 

Answer : We agree with your opinion and have removed “metabolism”, so the sentence was changed “ induces serious damaging of the bacterial membrane changing of the shape and activity